# Align and Adapt: Enhancing LLM Format Alignment and Knowledge Adaptation via Reverse Constraints Generation

## Abstract

Building effective LLM agents requires strong instruction-following capability in addition to domain knowledge. While human-annotated long-form QA (LFQA) datasets contain rich factual content, we find that directly fine-tuning on them degrades instruction-following performance, making it impractical to create domain-specific agents. Recent research on instruction-tuning has focused on augmenting existing instruction-tuning or conversational datasets to create complex instruction-tuning dataset, enabling LLMs to better handle fine-grained and nuanced instructions. While effective, these augmentation approaches risk distorting semantic meaning of the long-form QA datasets. We propose **REFER** (**RE**structure, **F**eature Extract, **R**everse constraint generation), a framework that transforms human-annotated long-form QA datasets into high-quality instruction-tuning datasets focused on verifiable constraints. **REFER** preserves the original semantics while integrates fine-grained format constraints into the dataset, enabling LLMs to improve instruction-following capability without sacrificing domain knowledge. Extensive evaluations on instruction-following benchmarks show that LLaMA-2-7B models fine-tuned with **REFER** exhibit stronger generalization in complex and multi-turn instruction following compared to both standard instruction-tuning and direct LFQA fine-tuning. REFER also emphasizes security and efficient where all the data augmentation is performed without external APIs, and supervised fine-tuning uses lightweight, reproducible LoRA adapters. Our results demonstrate that REFER enables the practical creation of domain-specific LLM agents with enhanced instruction-following capability which is something unattainable with naive LFQA fine-tuning.

## 1 Introduction

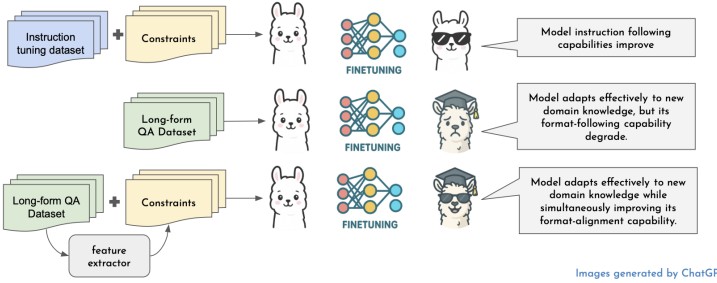

Figure 1: LLM Agent fine-tuning

The robust instruction-following abilities of LLMs have enabled human to use LLMs at areas that required precision and stability. IFEval (Zhou et al., 2023) is the first research that formalizes the evaluation of instruction-following when multiple constraints or sub-instructions are involved. IFEval (Zhou et al., 2023) has since inspired research into how prompts with multiple constraints can

improve a model's ability to follow nuanced instructions. Recently, a growing number of studies have begun to augment existing instruction-tuning dataset, such as Alpaca (Taori et al., 2023) and ShareGPT, to include additional fine-grained constraints. Format constraints are especially useful in real-world systems where output must follow fixed format such as number of words, capital letters and number of paragraphs.

Recent research such as Conifer (Sun et al., 2024) and UltraIF (An et al., 2025) have focused on improving complex-instruction following capabilities of smaller open-source LLMs by leverage the power of stronger LLMs to generate complex instruction-tuning dataset. Other studies such as From Complex to Simple (He et al., 2024a) and Verifiable Format Control (Wang et al., 2025) adapting contrastive learning approaches, where student models are prompted with complex task and their outputs are refined by stronger teacher models or rule-based method to form positive and negative training pairs. These data are the employed in reinforcement learning (Ouyang et al., 2022) to further improve the complex instruction following capability of student models. A primary drawback of using large language models to generate instruction-tuning dataset is the lack of factual grounding, as it is often impossible to verify the factuality of the generated content. Moreover, these approaches largely ignore the answer component of source datasets, instead generating responses solely from the LLM without reference to the original answers making them are unsuitable for transforming long-form QA datasets into complex instruction following tasks.

More recent research has leveraged the "reverse engineering" capabilities of large language models (LLMs) to construct high-quality instruction tuning datasets. Qi et al (Qi et al., 2025) utilized back translation to inject constraints into the question of existing instruction tuning dataset such as Alpaca (Taori et al., 2023), Evol-Instruct (Xu et al., 2025) to create complex instruction-tuning datasets. Pham et al. (Pham et al., 2024) employed back-translation techniques to generate instructions for the ChapterBreak (Sun et al., 2022) and Red Pajama (Weber et al., 2024) datasets. While these approaches preserve the original meaning of the datasets, several limitations remain when applied to human-annotated LFQA datasets. First, the phrasing of questions often lacks lexical diversity. Second, the answers tend to lack structural variety compared to those generated by large language models. Our proposed framework addresses these issues by enriching the diversity and structure of the source datasets before applying the "back-translation" approach for constraint generation.

Inspired by Verifiable Format Control (Wang et al., 2025) and Constraints Back Translation (Qi et al., 2025), we designed a dataset of constraints which can be added to existing long-form QA dataset without affecting its original meaning. We proposed a framework which leverage open-source LLM and NLP tools to augment existing long-form QA dataset. Our main goal is to propose a versatile framework that can transform existing domain specific dataset into instruction-tuning dataset, the augmented dataset can be used to effectively create domain specific LLM agent with enhanced instruction-following capabilities.

To demonstrate REFER's advantages, we compare the model fine-tuned with dataset augmented by REFER framework and model fine-tuned with dataset from other recent work in instruction-tuning. Our contributions are summarized as follows:

1. We show that fine-tuning large language models (LLMs) directly on human-annotated long-form QA dataset can degrade the model inherent instruction-following capability. This is due to the long-form QA not align with the model supervised fine-tuning objective and incoherence between questions and answers present in human annotated long-form QA.

2. We carefully curate a constraints dataset that contains various format constraints, which can be integrated into existing long-form QA datasets. These datasets are specifically designed to contain no semantic content and do not interfere with the original meaning of the source datasets.

3. We propose a framework that applies our custom constraints to long-form QA datasets, transforming them into instruction-tuning datasets focused on format alignment.

## 2 RELATED WORKS

**Format following.** Format alignment refers to a model's ability to respect structural, stylistic, or length-based constraints in its outputs. Earlier instruction-tuned models such as FLAN Wei et al. (2022) and Self-Instruct (Wang et al., 2023) mainly focused on task completion, without explicitly

enforcing output formats. Recent work (Wang et al., 2025) addresses this gap by generating format-constrained data with the rule-based method. However, these efforts rarely leverage long-form QA, which provides richer factual content, longer contexts, and greater domain diversity. Our work closes this gap by introducing reverse constraint generation to inject format constraints into LFQA data.

**Back Translation**. Instruction Induction (Honovich et al., 2023) demonstrates that LLMs are capable to infer underlying tasks and generate instructions from demonstrations. Recent researches (Pham et al., 2024; Qi et al., 2025) have leveraged the "reverse engineering" capabilities of large language models (LLMs) to construct high-quality instruction tuning datasets. Different from them, our work focus on transforming existing long-form QA dataset into complex instruction tuning datasets using both rule based and LLM based reverse constraint generation. Unlike datasets used by Qi et al., LFQA (e.g., Natural Questions (Kwiatkowski et al., 2019)) is rich in domain knowledge but structurally limited. Our proposed framework is capable of enhancing structural diversity while preserving semantics, making LFQA suitable for complex instruction-tuning.

**Long-form question answering**. LFQA is a challenging task in natural language processing, as it requires language models to generate coherent, knowledge-grounded answer that may spans hundreds of words. Unlike factoid QA (Stelmakh et al., 2022), where answers are usually labels or short sentence, long-form QA tasks (Fan et al., 2019; Kwiatkowski et al., 2019) challenge a model's in-context memory, reasoning ability and discourse-level coherence. Long-form question answering datasets provide rich, diverse and contextually grounded knowledge which serves as a useful resources to craft instruction following dataset. However, they are rarely leveraged for instruction tuning. With appropriate data augmentation techniques, it is possible to transform long-form QA data into effective fine-tuning dataset that enhance a model's format following capabilities while simultaneously adapting it to domain-specific content.

## 3 MOTIVATION

Building effective agents requires strong instruction-following capability in addition to domain knowledge. Human-annotated LFQA datasets such as Natural Questions (Kwiatkowski et al., 2019) and ELI5 (Fan et al., 2019) provide factually grounded knowledge and long-form answers with citations. However, directly fine-tuning LLMs on LFQA degrades instruction-following performance. This is because questions from such datasets often lack lexical diversity and answers are structurally limited. Moreover, question–answer coherence is often weak, since answers are extracted directly from snippets found on website or in literature rather than written for each query. We designed a framework that can mitigate these limitations and transform LFQA dataset into effective instruction-tuning dataset.

## 4 PROPOSED FRAMEWORK: REFER(REstructure, Feature Extract, REverse constraint generation)

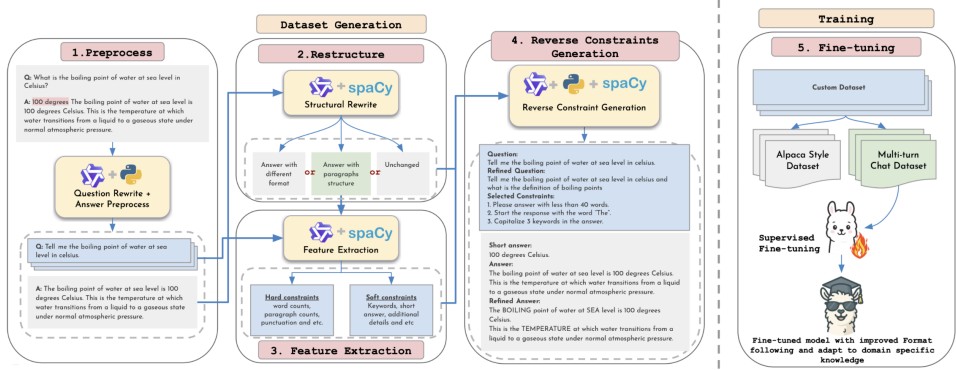

Figure 2: Proposed Framework

## 4.1 TASK DEFINITION

Our task is to transform LFQA data into instruction-tuning datasets where carefully designed constraints are integrated without altering factual meaning. Unlike prior work that only utilizes the questions from the source dataset, REFER leverages both the question and its long-form answer to preserve factual grounding. To further enhance model's performance, we re-frame the data into a multi-turn dialogue format. This design not only simulates realistic human–agent interactions but also helps models deepen their understanding of domain-specific knowledge while strengthening complex instruction following.

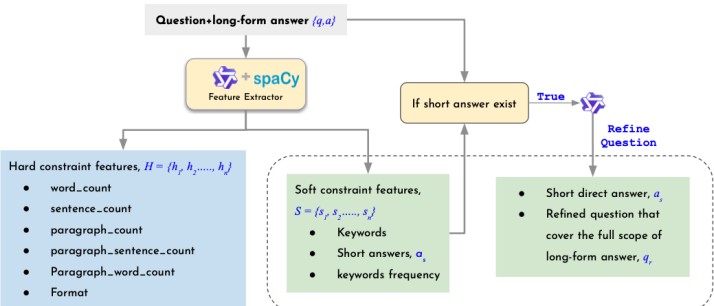

Figure 3: Feature Extraction

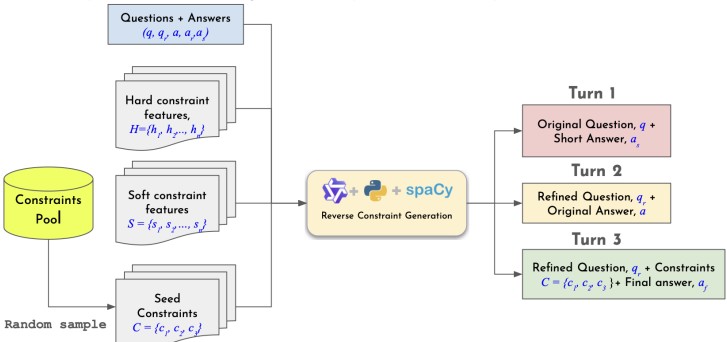

Figure 4: Multi-turn QA Dataset Generation

## 4.2 DATASET GENERATION PIPELINE

The REFER pipeline consists of four modules:

1. **Pre-processing.** Noise such as citations or page numbers is removed. The original questions are rewritten into diverse instruction ($q$) forms via few-shot prompting, enhancing lexical variety.

2. **Structural Rewrite.** Original answers ($a$) are reformatted into answers different structures ($a_r$) (e.g., multiple paragraphs, Markdown, JSONL) using LLM prompting. This increases structural diversity while preserving meaning.

3. **Feature Extraction.** We combine LLM specifically Qwen-3-32B (Yang et al., 2025) with SpaCy (Matthew Honnibal, 2020) and Python rule-based methods to extract features defined as $H = \{h_1, h_2, ..., h_n\}$ and $S = \{s_1, s_2, ..., s_n\}$ from the original answers. To address the incoherence between questions ($q$) and answers ($a$), we employ a few-shot prompting strategy to extract short, direct answers ($a_s$) from long-form responses. If the short answer exists within the long-form answer ($a$), it indicates that the question ($q$) is not sufficiently detailed to cover the full scope of the response. In such cases, we prompt

Qwen-3-32B to refine the question accordingly. The detailed feature extraction module is illustrated in Figure 3.

4. **Reverse Constraint Generation.** Constraints defined as $C = c_1, c_2, ..., c_n$ are derived from answer features ($H = \{h_1, h_2, ..., h_n\}$ and $S = \{s_1, s_2, ..., s_n\}$) (e.g., "limit to n paragraphs") and appended to the refined question ($q_r$) to form a more complex instruction. Beside reverse constraints, we also introduce lightweight structural edits to the answer (e.g., keyword highlighting, capitalization). Figure 4 illustrates how all components are integrated to create a multi-turn instruction-tuning dataset.

The multi-turn conversation dataset enable the models to learn format following constraints in QA settings. By combining short, direct QA pairs with more complex long-form QA within the same conversation, we can prevent the model from overgeneralizing toward producing long responses. This balanced dataset also helps the models learn to answer domain-specific questions at varying levels of granularity, further enhancing its responsiveness and adaptability.

## 5 EXPERIMENT SETUP

### 5.1 DATASET

We select Natural Questions (Kwiatkowski et al., 2019) created by Google as our source dataset. We use a cleaned version of Natural Questions (Thakur et al., 2021) which consists of textual content only, resulting in a smaller dataset of approximately 100,000 QA pairs. We sampled 40,000 QA pairs from the cleaned dataset based on the word counts of each answer, ensure an even distribution of answer length.

To evaluate the effectiveness of our framework in injecting new domain specific knowledge into LLMs, we utilize LFRQA dataset (Han et al., 2024). Unlike existing QA datasets such as Natural Questions (Kwiatkowski et al., 2019) and TriviaQA (Joshi et al., 2017), which primarily draw from Wikipedia or general web documents, LFRQA (Han et al., 2024) is explicitly designed to measure out-of-domain (OOD) performance. For our evaluation, we select the Finance and Lifestyle subsets in LFRQA. We construct two instruction-tuning datasets by combining the finance and Lifestyle subsets with general QA data from Natural Questions. Specifically

1. The first dataset includes 10,000 QA pairs from Natural Questions combined with 2,208 QA pairs from the Lifestyle subset.

2. The second dataset combines 10,000 QA pairs from Natural Questions with 3,612 QA pairs from the Finance subset.

Each of these combined datasets is then processed through our REFER framework to produce multi-turn conversational instruction datasets. We used four RTX 3090 GPUs with VLLM to enable distributed inference, which took about 20 hours to process 40,000 QA pairs.

### 5.2 BASELINES

We use the LLaMA-2-7B-Chat (Touvron et al., 2023) model as our pretrained model. The same training configuration is applied across all models. Specifically, we set the batch size to 8, the maximum sequence length to 4096, and the learning rate to $3 \times 10^{-5}$, using a cosine learning rate scheduler. Each model is trained for 2 epochs. To further improve training efficiency and reduce memory usage, we adopt the LoRA (Hu et al., 2021) fine-tuning method, with a LoRA rank of 16 and a LoRA alpha of 32. All models are trained on four RTX 3090 GPUs, with DeepSpeed utilized to enable distributed training. The total training time averages approximately 16 hours. The detailed dataset used for each model is shown in Table 1.

### 5.3 INSTRUCTION-FOLLOWING BENCHMARKS

**IFEval.** Instruction Following Evaluation (IFEval) (Zhou et al., 2023) is a widely adopted benchmark for assessing complex instruction-following. It contains 541 tasks, each task consists of 1 to 3 verifiable constraints which is objectively verified via rule-based scripts. IFEval provides two overall

| Models | Descriptions |
|---|---|
| Llama-2-7B ($Base$) | Base model. |
| Llama-2-7B ($LFQA$) | Llama-2-7B-Chat fine-tuned with original Natural Questions dataset reformatted into the ChatML structure. |
| Llama-2-7B ($REFER$) | Llama-2-7B-Chat fine-tuned with Natural Questions Dataset Refined with our REFER framework. |
| Conifer | Llama-2-7B-Chat fine-tuned with dataset proposed by Sun et al. (2024). |
| UltraIF | Llama-2-7B-Chat fine-tuned with dataset proposed by An et al. (2025). |
| ComToSim* | Llama-2-7B-Chat fine-tuned with dataset proposed by He et al. (2024a). |
| VFF* | Llama-2-7B-Chat fine-tuned with dataset proposed by Wang et al. (2025). |

Table 1: Details of baseline models. The reported results of models marked with * are extracted directly from the research papers.

scores: I-level (number of task where all constraints in a prompt satisfied) and C-level (percentage of individual constraints satisfied).

**Multi-IF.** Multi-IF (He et al., 2024b) extends IFEval to multi-turn and multilingual evaluation. It measures whether LLMs can consistently follow constraints across turns of conversations and transfer instruction-following capabilities to other languages when the instruction-tuning dataset is in English. Multi-IF poses greater challenge than typical single turn evaluation as LLMs often struggle to consistently adhere to instructions that were successfully executed in previous turns.

**LiveBench.** LiveBench (White et al., 2025) is a contamination free benchmark that refreshes every six months with newly created test cases from recent sources (e.g., arXiv papers, news). Similar to IFEval, LiveBench also scores answers automatically according to objective ground truth values which alleviate evaluator bias when LLMs are used as judges. We focus on its instruction-following subset for evaluation.

### 5.4 SYSTEM ANALYSIS

**Constraint Compatibility.** REFER generates constraints either by reverse generation from features extracted in answers, or apply light modifications to the original answer based on selected constraints. Constraint compatibility analysis allows us to understand whether the system is compatible with diverse datasets.

**Semantic Preservation.** To ensure that the addition of constraints into the dataset does not alter meaning of content, we compare original and refined answers using ROUGE-score Lin (2004). We report ROUGE-1, ROUGE-2, ROUGE-3, and ROUGE-L.

### 5.5 DOMAIN ADAPTATION ANALYSIS

we randomly sample 200 questions each from the Finance and Lifestyle subsets of LFRQA (Han et al., 2024). To prevent the models from relying on surface-level token patterns rather than true knowledge understanding, we use GPT 4o to paraphrase each question. We then prompt the fine-tuned models and base model to answer the rewritten questions and compare the generated responses against the original human-written ground truth answers.

**BERTScore.** We use BERTScore (Zhang et al., 2020) which compares contextual embeddings of answers generated by model and ground truth. This evaluation allows us to evaluate whether the fine-tuned models can retain and express domain-specific knowledge in a manner that is semantically consistent with the ground truth.

**LLM-based Evaluation.** Following recent work (Wei et al., 2025; Dubois et al., 2025), we use GPT 4o as a preference-based evaluator to compare fine-tuned and base model outputs. We present these two responses (a1 and a2) along with the corresponding ground truth answer (g) and the paraphrased question (q) to GPT 4o. The model is prompted to select the response that is more closely aligns with the ground truth.

# 6 RESULTS AND ANALYSIS

## 6.1 INSTRUCTION-FOLLOWING BENCHMARKS

**The main result of IFEval** (Zhou et al., 2023) is reported in Table 2 and Table 3. VFF proposed by Wang et al. (2025) reports only the I-level and C-level scores, whereas ComToSim introduced by He et al. (2024a) provides all the detailed scores of the benchmark. According to Wang et al. (2025), VFF is first trained via supervised fine-tuning, followed by reinforcement learning. Based on the research proposed by He et al. (2024a) Models labeled as $Generation$ are trained on dataset generated by GPT 3.5 turbo. Models labeled as $Discrimination$ are trained using dataset created with discrimination-based approach where the output of backbone models are refined by GPT 3.5 turbo.

Based on the results, both of our $REFER$ models consistently outperform the baselines in length generalization, highlighting the benefits of using long-form QA datasets. Our $REFER$ models achieve overall higher C-level scores compared to the baselines and perform better in 5 out of 9 constraint categories. Unlike the baselines that rely on instruction-tuning datasets as their source, our framework leverages LFQA datasets, which inherently lack any instruction-tuning function.

The $LFQA$ model shows significant performance degradation due to several factors. First, there is incoherence between the questions and answers in the dataset. Second, the lack of clear instructions and alignment between question and answer structures likely introduces a distribution shift from instruction-tuned objectives. These results demonstrate the importance of introducing format constraints into domain-specific datasets before fine-tuning LLMs.

To isolate the impact of our multi-turn design, we transformed the final round of our REFER dataset into a standard single-turn Alpaca-style instruction tuning format and fine-tuned the same model using the same hyper-parameters. The model fine-tuned with Alpaca dataset are labeled as Alpaca. This experiment shows that the model benefits from multi-turn QA fine-tuning.

| Models | ChangeCase | Combination | Content | Format | Keywords | Language |
|---|---|---|---|---|---|---|
| GPT 4* | 75.28 | 70.77 | 96.23 | 94.27 | 84.05 | 96.77 |
| ComToSim$_{Generation}$* | 41.57 | 15.38 | 71.70 | 70.70 | 53.37 | 58.06 |
| ComToSim$_{Discrimination}$* | 49.44 | 06.15 | **77.36** | 64.97 | 53.99 | 74.19 |
| VFF* | – | – | – | – | – | – |
| Conifer | 34.83 | 24.62 | **77.36** | 64.97 | **64.42** | 70.97 |
| UltraIF | 55.05 | **32.31** | 69.81 | 66.24 | 60.74 | **80.65** |
| Llama-2-7B ($Base$) | 32.61 | 07.71 | 81.19 | 63.70 | 62.63 | 41.98 |
| Llama-2-7B ($LFQA$) | 8.99 | 1.54 | 7.55 | 12.1 | 33.73 | 3.23 |
| Llama-2-7B ($Alpaca$) | 31.50 | 15.40 | 60.40 | 59.90 | 55.80 | 61.30 |
| Llama-2-7B ($REFER$) | **59.55** | 27.68 | 60.38 | **64.97** | 48.47 | 77.42 |
| Mistral-7B ($Base$) | **65.16** | **26.15** | 90.57 | **75.16** | 75.46 | **77.42** |
| Mistral-7B ($LFQA$) | 21.34 | 10.77 | 35.85 | 11.45 | 56.44 | 48.39 |
| Mistral-7B ($REFER$) | 64.03 | **26.15** | **90.57** | **75.16** | **76.07** | 74.19 |

Table 2: IFEval benchmark main results (Part A). The results marked with * are extracted directly from the research papers. We employ the strict metric from IFEval to calculate the accuracy scores.

| Models | Length | Punctuation | Startend | **I-Level** | **C-Level** |
|---|---|---|---|---|---|
| GPT 4* | 73.43 | 66.67 | 95.52 | 76.16 | 82.97 |
| ComToSim$_{Generation}$* | 27.97 | 9.09 | 56.72 | 34.01 | 46.16 |
| ComToSim$_{Discrimination}$* | 34.27 | 07.58 | 73.13 | 38.82 | 48.56 |
| VFF* | – | – | – | 40.48 | 54.08 |
| Conifer | 40.56 | 12.12 | 43.28 | 38.45 | 49.40 |
| UltraIF | 43.36 | 30.30 | 61.19 | **44.73** | 54.92 |
| Llama-2-7B ($Base$) | 39.26 | 13.65 | 49.37 | 30.12 | 40.37 |
| Llama-2-7B ($LFQA$) | 23.78 | 21.21 | 7.46 | 9.80 | 16.91 |
| Llama-2-7B ($Alpaca$) | 41.30 | 15.20 | 34.30 | 32.5 | 43.90 |
| Llama-2-7B ($REFER$) | **44.06** | **56.06** | **77.61** | 43.25 | **55.16** |
| Mistral-7B ($Base$) | 49.65 | 9.09 | 71.64 | 50.83 | 61.51 |
| Mistral-7B ($LFQA$) | 44.06 | **22.73** | 31.34 | 20.89 | 32.25 |
| Mistral-7B ($REFER$) | 50.34 | 9.09 | **74.63** | **51.20** | **61.75** |

Table 3: IFEval benchmark main results (Part B). The results marked with * are extracted directly from the research papers. We employ the strict metric from IFEval to calculate the accuracy scores.

**The main result of Multi-IF** (He et al., 2024b) is reported in Table 4. The results show trends similar to IFEval, where the $LFQA$ model exhibits significant degradation in instruction-following performance. The $REFER$ model performs noticeably better than Conifer and the base model, while performing slightly below the UltraIF model. Although the datasets generated by our REFER framework are exclusively in English, the results show that models fine-tuned using REFER exhibit cross-lingual generalization in instruction-following tasks. This suggests that with a well-designed instruction tuning strategy, models can transfer instruction-following capabilities to other languages even without fine-tuning on multilingual training data.

| Turn 1 | Average | Italian | Spanish | Hindi | Portuguese | English | French | Chinese | Russian |
|---|---|---|---|---|---|---|---|---|---|
| Conifer | 39.71 | 41.43 | 43.64 | 20.70 | 42.13 | 46.75 | **44.76** | 37.66 | 34.80 |
| UltraIF | **44.01** | **48.16** | 46.55 | **37.84** | **46.14** | 51.40 | 43.46 | **45.35** | 37.84 |
| Llama-2-7B ($Base$) | 33.77 | 36.74 | 35.41 | 16.93 | 38.67 | 42.57 | 36.02 | 29.91 | 26.33 |
| Llama-2-7B ($LFQA$) | 14.24 | 15.89 | 14.54 | 13.96 | 14.87 | 14.00 | 13.96 | 13.51 | 13.30 |
| Llama-2-7B ($alpaca$) | 32.25 | 35.83 | 34.66 | 15.35 | 35.00 | 42.94 | 33.16 | 26.51 | 25.61 |
| Llama-2-7B ($REFER$) | 42.39 | 43.98 | **46.73** | 19.43 | 41.74 | **52.60** | 44.10 | 40.48 | **41.77** |
| **Turn 2** | **Average** | **Italian** | **Spanish** | **Hindi** | **Portuguese** | **English** | **French** | **Chinese** | **Russian** |
| Conifer | 27.00 | 27.93 | 31.64 | 16.49 | 27.99 | 30.31 | 30.68 | 28.78 | 19.85 |
| UltraIF | **32.30** | **37.46** | **36.30** | **18.16** | **32.83** | **38.51** | 34.51 | **34.39** | **22.22** |
| Llama-2-7B ($Base$) | 26.47 | 29.46 | 30.89 | 11.63 | 28.95 | 34.32 | 29.58 | 24.37 | 16.27 |
| Llama-2-7B ($LFQA$) | 9.71 | 11.30 | 9.67 | 8.68 | 10.35 | 10.23 | 9.49 | 8.75 | 8.81 |
| Llama-2-7B ($alpaca$) | 23.90 | 26.65 | 28.81 | 12.08 | 23.63 | 31.42 | 25.02 | 22.09 | 15.58 |
| Llama-2-7B ($REFER$) | 30.33 | 33.87 | 35.66 | 12.73 | 30.20 | 38.27 | **34.86** | 30.29 | 20.99 |
| **Turn 3** | **Average** | **Italian** | **Spanish** | **Hindi** | **Portuguese** | **English** | **French** | **Chinese** | **Russian** |
| Conifer | 20.82 | 21.67 | 22.84 | 12.50 | 23.97 | 23.27 | 22.30 | 21.89 | 16.46 |
| UltraIF | **25.64** | **28.72** | **28.95** | **15.70** | **26.77** | 29.47 | 27.27 | **27.88** | 17.91 |
| Llama-2-7B ($Base$) | 21.53 | 23.17 | 24.10 | 11.30 | 23.54 | 27.29 | 23.10 | 20.19 | 15.02 |
| Llama-2-7B ($LFQA$) | 9.99 | 10.97 | 10.37 | 8.86 | 10.16 | 10.68 | 9.65 | 9.60 | 9.06 |
| Llama-2-7B ($alpaca$) | 19.44 | 20.50 | 22.04 | 11.28 | 20.72 | 24.77 | 20.18 | 17.88 | 13.95 |
| Llama-2-7B ($REFER$) | 25.24 | 27.22 | 28.51 | 12.16 | 26.61 | **31.68** | **28.19** | 24.38 | **18.29** |

Table 4: Detailed Multi-IF benchmark of Llama-2-7B models fine-tuned with different version of Natural Questions datasets.

**The main result of LiveBench** (He et al., 2024b) is reported in Table 5. As shown in Table 6, all fine-tuned models exhibit performance degradation. The $LFQA$ model shows the most significant decline in performance as expected. The UltraIF model also experiences noticeable degradation, while our REFER and Conifer models demonstrate comparatively smaller declines in performance. This trend can be attributed to the nature of LiveBench which leverages real-world data to construct its evaluation set, making it more challenging than other benchmarks. The UltraIF model which fine-tuned on synthetic datasets generated by the LLaMA-3.1-70B model, has limited exposure to real-world contexts. In contrast, the Conifer model which trained on the ShareGPT dataset generated by the more powerful GPT-4 Achiam et al. (2023), benefits from exposure to more up-to-date data. Our $REFER$ model, which utilizes a human-annotated dataset augmented with the Qwen-3-32B model, making it more robust in real-world scenarios.

| Models | Base Model | Paraphrase | Simplify | StoryGeneration | Summarize | Average |
|---|---|---|---|---|---|---|
| GPT-4o | GPT-4o | 62.67 | 67.75 | 66.25 | 63.1 | 64.94 |
| Mistral-7B ($Base$) | Mistral-7B-Instruct-v0.2 | 37.85 | **49.97** | 42.47 | **37.23** | **41.88** |
| Mistral-7B ($LFQA$) | Mistral-7B-Instruct-v0.2 | 24.23 | 20.62 | 23.15 | 23.52 | 22.88 |
| Mistral ($REFER$) | Mistral-7B-Instruct-v0.2 | **38.95** | 48.77 | **44.13** | 35.62 | 41.87 |
| Llama-2-7B ($Base$) | Llama-2-7B-chat | **28.67** | 47.05 | **37.22** | **33.67** | **36.65** |
| Llama-2-7B ($LFQA$) | Llama-2-7B-chat | 21.95 | 23.73 | 17.42 | 19.18 | 20.57 |
| Llama-2-7B ($REFER$) | Llama-2-7B-chat | 26.95 | 38.10 | 23.80 | 32.63 | 30.37 |

Table 5: LiveBench Benchmark Main Results.

## 6.2 SYSTEM ANALYSIS

**Constraint Compatibility.** Constraints are generated either by extracting features from answers (reverse generation) or by applying light modifications. Based on results shown in Table 8a, on

| Models | Base Model | Paraphrase | Simplify | Story Generation | Summarize | Average |
|---|---|---|---|---|---|---|
| Mistral-7B ($Base$) | Mistral-7B-Instruct-v0.2 | $0.00\%$ | $0.00\%$ | $0.00\%$ | $0.00\%$ | $0.00\%$ |
| Mistral-7B ($LFQA$) | Mistral-7B-Instruct-v0.2 | $-35.98\%$ | $-58.74\%$ | $-45.49\%$ | $-36.83\%$ | $-44.26\%$ |
| Mistral ($REFER$) | Mistral-7B-Instruct-v0.2 | $+2.91\%$ | $-2.40\%$ | $+3.91\%$ | $-4.32\%$ | $+0.03\%$ |
| Llama-2-7B ($Base$) | Llama-2-7B-chat | $0.00\%$ | $0.00\%$ | $0.00\%$ | $0.00\%$ | $0.00\%$ |
| Llama-2-7B ($LFQA$) | Llama-2-7B-chat | $-23.44\%$ | $-49.56\%$ | $-53.20\%$ | $-43.04\%$ | $-42.31\%$ |
| Llama-2-7B ($Alpaca$) | Llama-2-7B-chat | $-10.95\%$ | $-31.03\%$ | $-22.70\%$ | $-16.48\%$ | $-20.29\%$ |
| Llama-2-7B ($REFER$) | Llama-2-7B-chat | $-6.00\%$ | $-19.02\%$ | $-36.06\%$ | $-3.09\%$ | $-16.04\%$ |

Table 6: LiveBench Benchmarks (Percentage of performance gain).

| Models | BERTScore | | | GPT4o Evaluation | |
|---|---|---|---|---|---|
| | Precision | Recall | F1 | Align with $g$ | Score |
| Vanilla | 80.52 | 85.95 | 83.13 | 85/200 | 42.50 |
| Llama-2-7B ($lifestyle$) | **84.93** | **86.61** | **85.74** | **115/200** | **57.50** |
| Vanilla | 81.82 | 86.13 | 83.91 | 87/200 | 43.50 |
| Llama-2-7B ($fiqa$) | **87.22** | **87.32** | **87.25** | **113/200** | **56.50** |

Table 7: Domain Adaptation Evaluation: BERTScore and LLM Evaluation

40k QA pairs, REFER produced 75,582 constraints, with only 2.16% being incompatible with the dataset, showing the pool's versatility across diverse datasets such as Natural Questions.

**Semantic Preservation.** Since long-form QA answers are knowledge-rich, modifications must not alter factual meaning. Based on results shown in table 8b, we find high overlap between original and refined answers, confirming that REFER preserves semantic fidelity while integrating constraints.

| Dataset Analysis | Value |
|---|---|
| Dataset Size | 40,000 |
| Total Constraints | 75,582 |
| Incompatible Constraints | 1,635 |
| Incompatible Constraints Ratio | 2.16% |

| Evaluation Metrics | Precision | Recall | F1-score |
|---|---|---|---|
| ROUGE-1 | 0.97 | 0.96 | 0.96 |
| ROUGE-2 | 0.95 | 0.94 | 0.94 |
| ROUGE-3 | 0.94 | 0.93 | 0.93 |
| ROUGE-L | 0.97 | 0.96 | 0.96 |

(a) The ratio of selected constraints incompatible with the extracted features.

(b) ROUGE score of original answer compared to refined answer.

Table 8: System analysis.

## 6.3 DOMAIN ADAPTATION ANALYSIS

**BERTScore.** BERTScore is used to measure the semantic similarity between generated answers and human-written LFRQA references. As shown in Table 9, both fine-tuned models produce outputs that are more semantically aligned with the reference answers compared to the base model. This suggests that the fine-tuned models tend to generate more domain-specific and contextually appropriate responses.

**LLM-Based Evaluation.** We further conduct evaluation using GPT 4o for pairwise preference comparison, with LFRQA ground-truth as reference. Results in Table 9 indicate that higher number of outputs from fine-tuned models are preferred by the GPT 4o. This confirms that the fine-tuned models produce more accurate, direct, and domain-relevant answers. These findings demonstrate that REFER is effective in adapting models to new knowledge domains while improving answer quality and precision. To strictly prevent data leakage and memorization, we utilized GPT-4 to rewrite the evaluation set prompts and reference answers with different wording.

## 7 CONCLUSION

In this work, we propose REFER, a framework that transforms long-form QA datasets into high-quality instruction-tuning data with verifiable constraints. To create domain-specific LLM agents,

| Models | BERTScore | | | GPT4o Evaluation | |
|---|---|---|---|---|---|
| | Precision | Recall | F1 | Align with $g$ | Score |
| Vanilla | 80.52 | 85.95 | 83.13 | 85/200 | 42.50 |
| Llama-2-7B ($lifestyle$) | **84.93** | **86.61** | **85.74** | **115/200** | **57.50** |
| Vanilla | 81.82 | 86.13 | 83.91 | 87/200 | 43.50 |
| Llama-2-7B ($fiqa$) | **87.22** | **87.32** | **87.25** | **113/200** | **56.50** |

Table 9: Domain Adaptation Evaluation: BERTScore and LLM Evaluation

REFER systematically augments and re-frames human-annotated long-form QA datasets into multi-turn conversations, aligning large language models (LLMs) with format-following constraints while simultaneously adapting them to new domain knowledge. Our evaluations on IFEval (Zhou et al., 2023), Multi-IF (He et al., 2024b), and LiveBench (White et al., 2025) demonstrate that models fine-tuned with REFER not only maintain strong instruction-following capabilities but also generalize better in complex, multi-turn, and format-constrained scenarios. We contribute to the research community by open-sourcing the REFER framework and the associated constraint pool dataset to encourage other researchers to extend our work in future research in instruction tuning.

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

# A APPENDIX

## A.1 PROMPT SETTINGS

The prompt templates used in data augmentation are shown in Figures 5, 6, and 7.

---

**Question Rewrite Prompt Setting**

You are given a simple question and answer. Please rewrite the question into 5 new instructions with different vocabulary and writing style. Please maintain the original meaning of the question and only output the new instructions. You can refer to the example for guidance.

Example:
Question: when did the first episode of law and order air

Answer: Law & Order is an American police procedural and legal drama television series, created by Dick Wolf and part of the Law & Order franchise. It originally aired on NBC and, in syndication, on various cable networks. Law & Order premiered on September 13, 1990, and completed its 20th and final season on May 24, 2010. At the time of its cancellation, Law & Order was the longest-running crime drama on American primetime television. Its record of 20 seasons is a tie with Gunsmoke (1955–1975) for the longest-running live-action scripted American prime-time series with ongoing characters. Although it has fewer episodes than Gunsmoke, Law & Order ranks as the longest-running hour-long primetime TV series. Gunsmoke, for its first six seasons, was originally a half-hour program.

New instructions:

1. Determine the original broadcast date of the first episode of Law & Order.
2. Find out when Law & Order made its television debut.
3. What is the airdate of the pilot episode of the TV show Law & Order?
4. Identify the premiere date of the crime drama series Law & Order.
5. On what date did Law & Order first appear on television?

Now complete the task below:
Question: {question}
Answer: {answer}
New instructions:

---

Figure 5: Rewrite original question to increase the diversity of the dataset.

---

**Short Answer Extraction Prompt Setting**

You are given a question and a corresponding long-form answer. Your task is to extract the short, direct answer if it is explicitly present in the long-form response. The direct answer should be concise, without including additional explanation or context. If the question asks for a reason, explanation, description, or opinion, and the provided answer consists only of that explanatory content without a clearly extractable short answer, please output: [No direct answer]

Do not rewrite or summarize the long answer. Only extract the direct answer if it exists. Be strict in identifying clear direct answers.

Example 1:
Question: What is the boiling point of water at sea level in Celsius?
Answer: The boiling point of water at sea level is 100 degrees Celsius. This is the temperature at which water transitions from a liquid to a gaseous state under normal atmospheric pressure.
Output: 100 degrees Celsius

Example 2:
Question: Can you explain the process of mitosis?
Answer: Mitosis is a type of cell division that results in two daughter cells each having the same number and kind of chromosomes as the parent nucleus. The process consists of several stages including prophase, metaphase, anaphase, and telophase. It is essential for growth and tissue repair in multicellular organisms.
Output: [No direct answer]

Now complete the task below:
Question: {question}
Answer: {answer}

---

Figure 6: Extract short direct answer from long-form answer.

---

**Question Refine Prompt Setting**

You are given:

- Original question
- Short answer: A portion of the long-form answer that directly responds to the original question.
- Long-form answer: A more detailed response that includes both the short answer and additional relevant information.

Your task is to expand and improve the original question so that it better reflects the full scope of the long-form answer. Focus on incorporating aspects of the long-form answer that would be missing or unexpected if someone had only seen the original question and short answer. Only output the refined question.

Original Question: {question}
Short answer: {short_answer}
Long-form answer: {answer}
Refined question:

---

Figure 7: Refine original answer when short direct answer exist within the long-form answer. This stage of question refine aims to improve the coherence between the question and answer.

## A.2 CONSTRAINTS POOL EXAMPLE

Below are some examples of constraints from our constraint pool. The constraints used in our REFER system were written by humans and then rephrased into different versions with the same meaning using GPT 4o. Each constraint in the pool is rewritten into 3 to 4 variants with different wording and style to enhance the richness of the constraint pool.

```
{"category": "structure", "type": "length_constraint_less", "constraint":
↪    "The response should be fewer than {num} words."}
{"category": "structure", "type": "sentence_less", "constraint": "Keep
↪    the response to fewer than {num} sentences."}
{"category": "structure", "type": "sentence_more", "constraint":
↪    ["Provide a minimum of {num} sentences in your response.", "Please
↪    only response with one sentence."]}
{"category": "structure", "type": "sentence_word_specific", "constraint":
↪    "The {num_1} sentence should be at least {num_2} words."}
{"category": "caps", "type": "caps_no_caps", "constraint": "Do not use
↪    any uppercase letters in the response."}
{"category": "caps", "type": "caps_only_capital", "constraint":
↪    "Capitalize the first letter of every word in the response."}
{"category": "caps", "type": "caps_paragraph", "constraint": "Ensure the
↪    {num} paragraph is composed entirely of uppercase characters."}
{"category": "decoration", "type": "start_self", "modification": [],
↪    "constraint": "Start the response with \"{words}\"."}
{"category": "decoration", "type": "enclose", "modification":
↪    ["<start_of_answer> ", " <end_of_answer>"], "constraint": "Place the
↪    complete answer between \"<start_of_answer>\" and \"<end_of_answer>\"
↪    labels."}
{"category": "highlight", "type": "highlight", "modification": ["<",
↪    ">"], "constraint": ["Highlight {num} keywords in the response.
↪    Format each highlighted word as follows: <word>.", "Highlight 1
↪    keyword in the response. Format the highlighted word as follows:
↪    <word>."]}
{"category": "structure", "type": "paragraph_word_more", "constraint":
↪    "Each paragraph must have a minimum of {num} words."}
```

## A.3 MULTI-TURN INSTRUCTION-TUNING DATASET EXAMPLE

We use the same system prompt to fine-tune all models. The system prompt is shown below.

```
{"role": "system", "content": "You are an expert assistant who
↪    follows user instructions with precision. Always respond
↪    accurately and strictly obey all constraints on content, format,
↪    style, and wording."}
```

Each training instance consists of 2 to 3 rounds of conversation, with each round increasing in instruction complexity and the level of detail in the answers. Below are examples from the dataset generated by our REFER framework, using the Natural Questions dataset as the source.

**Example 1:**

- Round 1:

```
{"role": "user", "content": "What was the title of the debut
↪    studio album by the band Florence and the Machine?"}
{"role": "assistant", "content": "Florence and the Machine's
↪    debut studio album is titled *Lungs*."}
```

- Round 2:

```
{"role": "assistant", "content": "Florence and the Machine
↪  released their first extended play, A Lot of Love. A Lot
↪  of Blood, in March 2009. Their debut studio album, Lungs,
↪  was released in July 2009 through Island Records, reaching
↪  number one on the UK Albums Chart in January 2010. The
↪  album was subsequently certified quintuple platinum in the
↪  United Kingdom, quadruple platinum in Ireland and triple
↪  platinum in Australia. The album's lead single \"Kiss with
↪  a Fist\" peaked at number 51 on the UK Singles Chart. This
↪  was succeeded by the single \"Dog Days Are Over\", which
↪  reached number 23 in the UK and number 21 on the Billboard
↪  Hot 100 in the United States, and was certified quadruple
↪  platinum by the Recording Industry Association of America
↪  (RIAA). Third single \"Rabbit Heart (Raise It Up)\"
↪  reached number 12 in the UK and number 41 in Ireland.
↪  \"Drumming Song\" was released as the fourth single,
↪  charting at number 54 in the UK. \"You've Got the Love\",
↪  a cover of The Source's song of the same name, peaked at
↪  number five in the UK and number nine in Australia. A
↪  performance at the 2010 BRIT Awards on 17 February 2010
↪  saw the Dizzee Rascal-assisted mash-up \"You Got the
↪  Dirtee Love\" debut at number two in the UK. The album's
↪  sixth and final single, \"Cosmic Love\", peaked at number
↪  51 in the UK and number three in Ireland."}
```

- Round 3:

```
{"role": "user", "content": "What was the title of Florence
↪  and the Machine's debut studio album, and what were some
↪  of its notable chart achievements and singles? Ensure the
↪  response contains the terms \"discography\", \"lungs\"."}
{"role": "assistant", "content": "Florence and the Machine
↪  released their first extended play, A Lot of Love. A Lot
↪  of Blood, in March 2009. Their debut studio album, Lungs,
↪  was released in July 2009 through Island Records, reaching
↪  number one on the UK Albums Chart in January 2010. The
↪  album was subsequently certified quintuple platinum in the
↪  United Kingdom, quadruple platinum in Ireland and triple
↪  platinum in Australia.\n\nThe album's lead single \"Kiss
↪  with a Fist\" peaked at number 51 on the UK Singles Chart.
↪  This was succeeded by the single \"Dog Days Are Over\",
↪  which reached number 23 in the UK and number 21 on the
↪  Billboard Hot 100 in the United States, and was certified
↪  quadruple platinum by the Recording Industry Association
↪  of America (RIAA). Third single \"Rabbit Heart (Raise It
↪  Up)\" reached number 12 in the UK and number 41 in
↪  Ireland. \"Drumming Song\" was released as the fourth
↪  single, charting at number 54 in the UK.\n\n\"You've Got
↪  the Love\", a cover of The Source's song of the same name,
↪  peaked at number five in the UK and number nine in
↪  Australia. A performance at the 2010 BRIT Awards on 17
↪  February 2010 saw the Dizzee Rascal-assisted mash-up \"You
↪  Got the Dirtee Love\" debut at number two in the UK. The
↪  album's sixth and final single, \"Cosmic Love\", peaked at
↪  number 51 in the UK and number three in Ireland."}
```

**Example 2:**

- Round 1:

```
{"role": "user", "content": "Identify the name of the actress
↪  that brought the character Teddy Altman to life on Grey's
↪  Anatomy."}
{"role": "assistant", "content": "Kim Raver played Teddy
↪  Altman on Grey's Anatomy."}
```

- Round 2:

```
        {"role": "user", "content": "Identify the actress who
        ↪   portrayed the character Teddy Altman on Grey's Anatomy,
        ↪   and provide some examples of her other notable television
        ↪   roles."},{"role": "assistant", "content": "Kim Raver
        ↪   Kimberly Jayne \"Kim\" Raver (born October 15, 1969) is an
        ↪   American actress. She is best known for television roles
        ↪   as Kim Zambrano on Third Watch, Audrey Raines on 24 and
        ↪   Teddy Altman on ABC's medical drama Grey's Anatomy."}
```

- Round 3:

```
        {"role": "user", "content": "Identify the actress who
        ↪   portrayed the character Teddy Altman on Grey's Anatomy,
        ↪   and provide some examples of her other notable television
        ↪   roles. Ensure the answer contains no fewer than 2
        ↪   sentences. Include the \"actress\" exactly once in your
        ↪   answer. Start the response with the exact phrase
        ↪   \"Welcome!\"."}
        {"role": "assistant", "content": "Welcome! Kim Raver Kimberly
        ↪   Jayne \"Kim\" Raver (born October 15, 1969) is an American
        ↪   actress. She is best known for television roles as Kim
        ↪   Zambrano on Third Watch, Audrey Raines on 24 and Teddy
        ↪   Altman on ABC's medical drama Grey's Anatomy."}
```

**Example 3:**

- Round 1:

```
        {"role": "user", "content": "What team did Kobe Bryant face
        ↪   when he recorded his highest-scoring game of 81 points?"}
        {"role": "assistant", "content": "Kobe Bryant scored 81 points
        ↪   against the Toronto Raptors in 2006."}
```

- Round 2:

```
        {"role": "user", "content": "What team did Kobe Bryant face
        ↪   when he scored his career-high 81 points, and how does
        ↪   this game fit into the broader context of his impressive
        ↪   list of high-scoring performances in NBA history?"}
        {"role": "assistant", "content": "List of 40-plus point games
        ↪   by Kobe Bryant. The number of forty-plus point games
        ↪   players accumulate over their careers is often reported in
        ↪   media. Bryant has played 135 games in which he has scored
        ↪   40 or more points; of these, 6 were 60-plus point games
        ↪   and 26 were 50-plus point games. He is third behind Wilt
        ↪   Chamberlain and Michael Jordan, who scored 40 or more in
        ↪   284 and 211 games, respectively. In 2003, Bryant scored 40
        ↪   points or more in nine consecutive games, tying Jordan,
        ↪   who accomplished the same feat in the 1986{87 season. The
        ↪   only player with longer streaks of 40 or more is
        ↪   Chamberlain, who had 14 consecutive games twice in the
        ↪   1961{62 season and 10 consecutive games in the 1962{63
        ↪   season. In 2006, Bryant scored a career-high 81 points
        ↪   against the Toronto Raptors. It was the second-highest
        ↪   number of points scored in a game in NBA history, behind
        ↪   only Chamberlain's 100-point performance in 1962. In 2007,
        ↪   Bryant scored 50 points or more in four consecutive games;
        ↪   this accomplishment is fifth in NBA history behind streaks
        ↪   by Chamberlain, who had 50 or more in seven, six and five
        ↪   (twice) consecutive games in the 1961{62 season. Bryant
        ↪   has also played 12 playoff games in which he has scored
        ↪   forty or more points. Out of the 134 games, 21 resulted in
        ↪   Bryant notching a double-double and 42 resulted in losses.
        ↪   Bryant became the oldest player to score 60+ points (60)
        ↪   in his final game on April 13, 2016."}
```

- Round 3:

```
{"role": "user", "content": "What team did Kobe Bryant face
↪  when he scored his career-high 81 points, and how does
↪  this game fit into the broader context of his impressive
↪  list of high-scoring performances in NBA history? Ensure
↪  each paragraph contains a minimum of 4 sentences. Ensure
↪  the term \"double-double\" appears once in your reply."}
{"role": "assistant", "content": "List of 40-plus point games
↪  by Kobe Bryant. The number of forty-plus point games
↪  players accumulate over their careers is often reported in
↪  media. Bryant has played 135 games in which he has scored
↪  40 or more points; of these, 6 were 60-plus point games
↪  and 26 were 50-plus point games. He is third behind Wilt
↪  Chamberlain and Michael Jordan, who scored 40 or more in
↪  284 and 211 games, respectively. In 2003, Bryant scored 40
↪  points or more in nine consecutive games, tying Jordan,
↪  who accomplished the same feat in the 1986{87 season. The
↪  only player with longer streaks of 40 or more is
↪  Chamberlain, who had 14 consecutive games twice in the
↪  1961{62 season and 10 consecutive games in the 1962{63
↪  season.\n\nIn 2006, Bryant scored a career-high 81 points
↪  against the Toronto Raptors. It was the second-highest
↪  number of points scored in a game in NBA history, behind
↪  only Chamberlain's 100-point performance in 1962. In 2007,
↪  Bryant scored 50 points or more in four consecutive games;
↪  this accomplishment is fifth in NBA history behind streaks
↪  by Chamberlain, who had 50 or more in seven, six and five
↪  (twice) consecutive games in the 1961{62 season. Bryant
↪  has also played 12 playoff games in which he has scored
↪  forty or more points. Out of the 134 games, 21 resulted in
↪  Bryant notching a double-double and 42 resulted in losses.
↪  Bryant became the oldest player to score 60+ points (60)
↪  in his final game on April 13, 2016."}
```

## A.4 THE USE OF LARGE LANGUAGE MODELS (LLMS)

Large Language Models (LLMs), specifically GPT-4o Achiam et al. (2023), are used in three major areas of this paper.

- We create the constraint pools shown in Section A.2 with the assistance of the GPT-4o model Achiam et al. (2023). First, we construct a dataset of seed constraints, each uniquely written by humans. We then use GPT-4o to expand this dataset by rewriting each constraint into 3–4 variants with different wording and style while preserving the original meaning. This increases the diversity of the constraints and helps the fine-tuned model acquire new skills without overfitting.

- We use the GPT-4o model to refine and correct any grammatical errors in our prompt settings, as shown in Section A.1.

- We use the GPT-4o model to improve the clarity and fluency of the paper's writing.

The use of open-source model, specifically Qwen-3-32B model is used in the REFER framework. The details are shown in Section 4.

