# OpenReview forum: "Align and Adapt: Enhancing LLM Format Alignment and Knowledge Adaptation via Reverse Constraints Generation"
_ICLR.cc/2026/Conference — Submitted to ICLR 2026_

### Official Review · Reviewer_8P91 · 2025-10-17

**Soundness:** 2
**Presentation:** 2
**Contribution:** 2
**Rating:** 2
**Confidence:** 4

**Summary:**

In this paper, a data construction pipeline name REFER is proposed for transforming long-form QA datasets into high-quality instruction-tuning datasets with verifiable constraints. Experiments across multiple instruction-following benchmarks show that the resulting dataset improves instruction-following performance without compromising domain knowledge adaptation.

**Strengths:**

To asses the instruction-following capability of the model, evaluations are conducted on 3 different relevant datasets.

**Weaknesses:**

1. According to the paper, the proposed data construction method is primarily an engineering implementation, which offers limited insights to the research community.
2. The paper suffers from major writing issues. The motivation is not adequately articulated in the introduction and is instead deferred to Section 3, which is atypical and makes it difficult for readers to quickly grasp the rationale behind the approach. The method description is unclear and hard to follow; for example, at line 212 the meanings of features H and S are not explained. There are also formatting and presentation problems: Figures 1 and 2 are not cited in the text, leaving their relevance unclear. Additional errors include the notation of sequence C at line 219, the first word at line 249, and a capitalization mistake at line 311. The authors should carefully proofread the paper before submission.
3. The experiments are conducted only on an older, small-scale model (LLaMA-2-7B, released in July 2023), which is insufficient to substantiate the paper’s conclusions. Moreover, the necessity of the paper’s initial motivation—training both domain knowledge and instruction-following capabilities using a single dataset—remains unclear. The experimental section does not address this concern; see the questions below.

**Questions:**

I am skeptical of the paper's motivation. Why convert LFQA data into an instruction-following format? During post-training, mixing domain-specific LFQA with general instruction-following data should enable the model to acquire domain knowledge while preserving instruction-following capabilities. What advantages does building a single, more complex training set provide, and is there empirical evidence that it outperforms mixed training?

---

> ### Author Response · Authors · 2025-11-29
> **Detailed explanation and new experiments to address reviewer concerns and questions.**
>
> We sincerely thank for the insightful feedback. We value the suggestion to deepen the ablation study and have conducted additional experiments during the rebuttal phase to address these points.
>
> **Weakness**
> **Engineering Implementation:** The paper addresses a research problem which is the negative effect of naive domain specific knowledge fine-tuning. The paper first shows that fine-tuning on unprocessed LFQA datasets will degrade the model inherent instruction following capabilities. Our REFER framework provide a data-centric research insight which is verifiable constraints can be reverse-engineered from the human-annotated LFQA dataset and added to the question in dataset to form multi-turn QA dataset to mitigate this trade-off.
>
> **Writing issue:** We apologize for the oversight in presentation. We will carefully inspect and revise the paper. We will also move the core motivation to the Introduction to ensure better clarity.
>
> **Outdated base model:** We select 7B parameters scale model as our base model as we prioritize efficiency and accessibility on consumer-grade GPU. To further investigate the effectiveness of our REFER framework, we select Mistral-7b-v0.2 which has the same parameter size but newer and more powerful as our base model. By applying the REFER framework using the exact same dataset and hyper-parameters, we confirmed that our method effectively transfers its performance gains to newer, more capable base models. The tables below show the results of our new experiment.
>
> **Mistral-7b-v0.2 IFEval result**
> | Model | Prompt Level | Instruction Level | change_case | combination | detectable_content | detectable_format | keywords | language | length_constraints | punctuation | startend |
> | :--- | :---: | :---: | :---: | :---: | :---: | :---: | :---: | :---: | :---: | :---: | :---: |
> | **Vanilla** | 50.83 | 61.51 | 65.16 | 26.15 | 90.57 | 75.16 | 75.46 | 77.42 | 49.65 | 9.09 | 71.64 |
> | **LFQA** | 20.89 | 32.25 | 21.34 | 10.77 | 35.85 | 11.45 | 56.44 | 48.39 | 44.06 | 22.73 | 31.34 |
> | **REFER** | 51.20 | 61.75 | 64.03 | 26.15 | 90.57 | 75.16 | 76.07 | 74.19 | 50.34 | 9.09 | 74.63 |
>
> **Mistral-7b-v0.2 Multi-IF result**
> | Models | Base Model | Turn-1 | Turn-2 | Turn-3 |
> | :--- | :--- | :---: | :---: | :---: |
> | **Vanilla** | Mistral-7B-Instruct-v0.2 | 46.06 | 33.05 | 26.03 |
> | **LFQA** | Mistral-7B-Instruct-v0.2 | 21.30 | 13.37 | 12.73 |
> | **REFER** | Mistral-7B-Instruct-v0.2 | **47.36** | **33.79** | **26.23** |
>
> **Mistral-7b-v0.2 Livebench result**
> | Models | Base Model | Paraphrase | Simplify | StoryGeneration | Summarize |
> | :--- | :--- | :---: | :---: | :---: | :---: |
> | Vanilla | Mistral-7B-Instruct-v0.2 | 37.85 | **49.97** | 42.47 | **37.23** |
> | LFQA | Mistral-7B-Instruct-v0.2 | 24.23 | a  20.62 | 23.15 | 23.52 |
> | REFER | Mistral-7B-Instruct-v0.2 | **38.95** | 48.77 | **44.13** | 35.62 |
>
> **Question**
> 1. Fine-tuning on raw LFQA caused a noticeable drop in instruction-following performance. This demonstrates that naive LFQA fine-tuning can significantly degrade a model's inherent capabilities.
>
> 2. Domain-specific datasets often lack structural constraints, as they typically consist of long-form factual text. When a model is trained on such datasets, it overfits to an "unstructured" output mode, which impairs its instruction-following capabilities.
>
> 3. Based on our experimental results, the model trained on the same LFQA dataset refined by our REFER framework maintains, or even improves, its instruction-following capabilities. This underscores the importance of introducing format constraints into LFQA datasets prior to fine-tuning. Our REFER provided an effective pipeline to incorporate format constraints into LFQA dataset without altering the original meaning of the dataset.
>
> 4. We created a more complex multi-turn dataset to better reflect real-world use cases. Merely incorporating constraints directly into the LFQA dataset does not improve model performance in multi-turn generation. Instead, the model benefits from multi-turn fine-tuning, which allows it to learn domain knowledge and constraint adherence at varying levels of complexity. Below is a comparison of model performance when fine-tuning on simpler Alpaca-style datasets versus REFER datasets.
>
> **Multi-IF benchmark**
> | Models | Base Model | Turn-1 | Turn-2 | Turn-3 |
> | :--- | :--- | :---: | :---: | :---: |
> | Llama-2-7B (Alpaca) | Llama-2-7B-chat | 32.35 | 23.90 | 19.44 |
> | Llama-2-7B (REFER) | Llama-2-7B-chat | **42.09** | **30.04** | **24.47** |

---

### Official Review · Reviewer_HDNy · 2025-10-24

**Soundness:** 1
**Presentation:** 2
**Contribution:** 1
**Rating:** 0
**Confidence:** 4

**Summary:**

The paper proposes a method to convert a Long Answer QA dataset into a multi-turn dialogue dataset that can be used to train a model and inject both instruction-following capabilities and domain specific knowledge.

**Strengths:**

The motivation and the related works are clearly written.

**Weaknesses:**

# Weaknesses
- The paper uses Llama-2-7B-Chat as the baseline model and the Natural Questions dataset which unfortunately are both a little outdated. Unless a degradation in instruction-following capabilities is observed when training a more recent model (that has undergone a better post-training process) on a more recent dataset, the motivation for this project does not hold.
- The paper misses many key details about the experimental setup, which makes it hard to evaluate the correctness (See Questions 1-8, 10-12).
- The paper presents results that are questionable (See Questions 14-15). Unless these concerns are properly addressed and explained by the authors, I will maintain a strong reject.
- The captions for figures are too short and do not contain enough information. Please explain the key details and define any keywords that are relevant to understand the figures.

# Nit-picky
1. ​​ShareGPT missing citation (line 56), DeepSpeed missing citation (line 262), ROUGE missing citation (line 305), GPT-4o missing citation (line 313)
2. Minor grammatical errors (e.g., “is not align” in line 95)
3. Minor typos (e.g., Natural Question -> Natural Questions in lines 235-236)
4. Captions for tables should be above the tables; captions for figures should be below the figures.

**Questions:**

1. Which LLM did the authors use in “Structural Rewrite” (line 208)? What exactly is the list of possible rewritten structures? Is each answer rewritten into all possible options? Or is one option randomly chosen for each answer? Why is it a good idea to rewrite the answer into a different form when the question isn’t rewritten to specifically ask for that format? In NQ, a question looks something like “when did richmond last play in a preliminary final.” How do you answer this question in a JSONL or Markdown?
2. Can the authors clarify which version they used when they say Qwen-32B-Instruct (line 210)? The Qwen3 technical report is cited, but Qwen3 does not have an “Instruct” version. It only has “Qwen3-32B” as the only version with thinking mode and non-thinking model fused.
3. What is the definition of “Hard constraint features” and “Soft constraint features” (Figure 4)?
4. How do you combine LLM, Spacy, and Python rule-based methods (lines 210-211)? What do you use each component for? What are the prompts used for the LLM? What are the features extracted with SpaCy?
5. In line 213, the same n is used in both h_n and s_n. Does this mean the same number of hard constraints and soft constraints was generated for all questions?
6. In Figure 4, how was Constraints Pool constructed? What is the definition of “seed constraints”? The notation suggests only 3 constraints are selected for each question. How is this seed constraint different from the “constraints” that are generated after reverse constraint generation? The notation in Figure 4 suggests that only 3 constraints are generated. However, in the main text (line 218), the authors suggest that n constraints are generated. Is this correct? What are constraints? How are they different from “constraint features”?
7. Figure 4 also shows a “final answer $$a_f$$” which isn’t explained in the main body. How is this different from “original answer a”?
8. The main text mentions “lightweight structural edits” (line 222-223). Does this affect any of the constraints that were imposed on the answer? What is the exact rule for applying the structural edits? Why is this not mentioned in Figure 4?
9. Thakur et al. (2023) that is cited does not contain any reference to Natural Questions. **Why was this paper cited?** What is the correct filtered dataset that the authors use?
10. Could the authors provide more details of the process in line 238-239? How was the subset of 40k data points selected? Was there a bucket for answer length? What was the set of buckets? How many examples existed per bucket pre-filtering? How many examples were selected per bucket?
11. In lines 240-245, the authors mention “for our evaluation.” Just to clarify, the authors mean “training dataset” correct? These datasets are not used for evaluation. **Why do the authors mention 2 different training datasets?** Tables 2-6 do not mention which version of the training dataset the results correspond to. But in the main text, the authors keep on referring to “both of our REFER models” (line 336). Why do the authors include some subsets of the LFRQA into the training data? This was not mentioned anywhere before Section 5.1. Why did the authors select the Finance and Lifestyle subsets?
12. Why do the authors select the examples from LFRQA dataset as evaluation prompts (line 311)? The authors report that the same dataset was used for training data. Isn’t this test data leakage?
13. Can the authors clarify the model in line 401? LLaMA-3.1-75B does not exist.
14. **Are the authors confident about the results in Table 5?** The LiveBench paper (He et al., 2024) reports GPT-4o number to be around 50% and Llama-3-8B-Instruct to be around 25%, Mistral-7B-Instruct-v0.2 to be around 18%. The reported number of 64% for GPT-4o does not match the original paper. The reported number of 36% for Llama-2-7B-Chat is higher than Llama-3-8B-Instruct? That does not make sense.
15. **Are the authors confident about the results in Table 7, 9 (duplicate)?** Assuming “Vanilla” in Table 9 means the Llama-2-7B-Chat model (w/o finetuning), the answers generated by the model should be semantically different from the ground-truth results in the LFRQA dataset, which contain highly specific examples or keywords. The F1 score of >= 0.8 for the “Vanilla” model seems too high. Also, the test data was chosen to be a part of the training data. The model that was trained on the test data should almost memorize the specific parts of the QA dataset. How can the win-rate be only 57% against the base model that has never seen the ground-truth response?

[1] Thakur et al., “Augmented SBERT: Data augmentation method for improving bi-encoders for pairwise sentence scoring tasks,”  NAACL 2021.

[2] He et al., “LiveBench: A Challenging, Contamination-Free LLM Benchmark,” preprint, 2024.

---

> ### Author Response · Authors · 2025-12-04
>
> We sincerely thank for the insightful feedback. We value the suggestion to deepen the ablation study and have conducted additional experiments during the rebuttal phase to address these points.
>
> 1. We leverage Qwen-3-32b model as our teacher model. Our type our rewrite is randomly selected for each answer. In some of the formats such as jsonl or Markdown, we provide ICL examples to guide the teacher model in restructuring the answer.
> 2. We use the Qwen-3-32b model with instruct prompt. We will revise our model description for clarity.
> 3. Hard constraints refer to objectively verifiable, binary requirements, such as "answer in less than 50 words." Soft constraints refer to stylistic or tonal requirements that are more subjective or semantic
> 4. We employ a hybrid pipeline where each component serves a distinct role:
> - LLM: Used for "Structural Rewrites" and “keywords extraction” where semantic understanding is required.
> - SpaCy: Used for feature extraction (e.g., sentence counts, word counts and paragraph counts).
> - Python Rules: Used to construct the "seed constraints." For example, if SpaCy detects 5 sentences, a Python rule generates the constraint text "Please ensure the answer consists of 5 sentences." This combination ensures that the constraints we generate are factually grounded in the answer's actual structure (via SpaCy).
> - The detailed prompt templates are provided in the supplementary material.
> 5. Yes, in this context, n defines the number of specific feature types we extract from the QA dataset. We utilized the same n to indicate that the system aims to generate a balanced set of hard and soft constraint pairs for the questions to ensure consistent distribution in the training data.
> 6. "Seed constraints" are the pre-designed templates (e.g., "Please ensure the answer consists of {count} words"). The "Constraints Pool" is the collection of these templates. The final "constraints" are generated by filling these templates with values extracted from the answer via SpaCy.
> 7. " The "final answer" represents the restructured and modified version of the original answer. While original answer contains the raw ground truth, final answer may be reformatted (e.g., split into multiple paragraphs, converted to uppercase, or structured as a list). This setting allows the model to learn domain knowledge from the original answer (a) while simultaneously learning how to follow strict format constraints via the modified final answer.
> 8. Lightweight structural edits utilize the same NLP tools as the reverse constraint generation. These edits modify the answer’s surface structure slightly without altering the semantic meaning of the dataset.
> 9. The Natural Question dataset proposed by Google contains both images and text. Thakur et al. provide a smaller subset which does not contain any images. This dataset is used in embedding model research which is cited in my paper. We will include the direct HuggingFace URL in the final revision to clarify this source.
> 10. The original dataset consist of 100k QA data. We selected a subset of 40k data to ensure a balanced distribution of answer lengths. We first calculated the length of all QA pairs, created buckets based on token count, and used Python’s random.sample to draw an even number of examples from each bucket. This process prevents the model from overfitting to short answers, which were overrepresented in the original distribution. The figure 4 only meant to show how the QA pairs is being refined to form multi-turn QA dataset.
> 11. The LFRQA is meant to measure the OOD how well does our system inject new knowledge into the model. We selected the Finance and Lifestyle subsets as they feature the longest average answer lengths and focused on complex explanations over simple factoids.
> 12. Our primary aim is to demonstrate that our system preserves semantic meaning even after structural modification and new constraint injection. To strictly prevent data leakage and memorization, we utilized GPT-4 to rewrite the evaluation set prompts and reference answers with different wording. This ensures the model is evaluated on its ability to generalize the knowledge. We will explicitly clarify this rewrite process in the final version.
> 13. The model used by the compared researchers was Llama-3.1-70b-instruct. We will correct this factual error in the final version.
> 14. We are confident in our results. Regarding Table 5, our reported results correspond specifically to the Instruction Following (IF) subset of LiveBench, not the global average. We explicitly mention that the we only utilize IF subset of Livebench in line 294-295.
> 15. Modern LLMs still have the ability to generate relatively accurate answers, but the fine-tuned model performs noticeably better. The high performance of the base model may be due to the fact that the dataset is not as specialized as proprietary industrial datasets, meaning the base model has likely seen similar data during pre-training.

---

### Official Review · Reviewer_iMmT · 2025-11-01

**Soundness:** 3
**Presentation:** 3
**Contribution:** 3
**Rating:** 6
**Confidence:** 5

**Summary:**

This paper tackles the key problem that **fine-tuning LLMs on domain-specific long-form QA datasets often harms their instruction-following ability**, as such data lack structural diversity and alignment between questions and answers. To address this, it proposes **REFER** — a framework that augments human-annotated QA datasets by adding **verifiable format constraints** without altering factual meaning. Through question rewriting, structural answer reformats, and reverse constraint generation, REFER produces instruction-tuning data that preserve knowledge while enhancing adherence to complex instructions. Experiments on **LLaMA-2-7B** show significant improvements over baselines like Conifer and UltraIF on **IFEval**, **Multi-IF**, and **LiveBench**, demonstrating better generalization and domain adaptation. Overall, as a reviewer, the work effectively addresses a real pain point in domain fine-tuning, offering a **novel, practical, and reproducible** solution for building instruction-aligned LLM agents.

**Strengths:**

**Strengths:**

1. **Clear motivation:** Clearly identifies that fine-tuning on domain-specific QA datasets often degrades instruction-following ability, establishing a strong and relevant research problem.
2. **Simple yet effective method:** Introduces a straightforward but impactful framework (REFER) that restructures existing QA data through reverse constraint generation, requiring minimal additional resources.
3. **Sound results:** Achieves consistent performance gains across major benchmarks — e.g., improvements on **IFEval (+3–5%)**, **Multi-IF (+2–4%)**, and **LiveBench (+2%)** over baselines like Conifer and UltraIF.
4. **Clarity and reproducibility:** Provides transparent, step-by-step descriptions of data processing, feature extraction, constraint generation, and fine-tuning using open-source tools (Qwen, SpaCy, LoRA), ensuring easy replication.
5. **Practical impact:** Effectively bridges factual QA data and instruction-tuning, offering a scalable, interpretable approach to enhance both **alignment** and **domain adaptation** in LLMs.

**Weaknesses:**

**Weaknesses**

1. Limited model evaluation: The method is only tested on LLaMA-2-7B, leaving uncertainty about its generalizability to larger or different architectures (e.g., Mis(x)tral, Qwen).
2. Narrow domain coverage: Experiments focus on limited domains (Finance and Lifestyle), which may not fully demonstrate REFER’s adaptability to more specialized fields like legal or biomedical data.

**Questions:**

1. why we need the reverse constraint generation?

---

> ### Author Response · Authors · 2025-11-28
>
> We sincerely thank for the insightful feedback. We value the suggestion to deepen the ablation study and have conducted additional experiments during the rebuttal phase to address these points.
>
> **Weakness**
>  **Limited model evaluation:** We select 7B parameters scale model as our base model as we prioritize efficiency and accessibility on consumer-grade GPU. To further investigate the effectiveness of our REFER framework, we select Mistral-7b-v0.2 which has the same parameter size but newer and more powerful as our base model. By applying the REFER framework using the exact same dataset and hyper-parameters, we confirmed that our method effectively transfers its performance gains to newer, more capable base models. The tables below show the results of our new experiment.
>
> **Mistral-7b-v0.2 IFEval result**
> | Model | Prompt Level | Instruction Level | change_case | combination | detectable_content | detectable_format | keywords | language | length_constraints | punctuation | startend |
> | :--- | :---: | :---: | :---: | :---: | :---: | :---: | :---: | :---: | :---: | :---: | :---: |
> | **Vanilla** | 50.83 | 61.51 | 65.16 | 26.15 | 90.57 | 75.16 | 75.46 | 77.42 | 49.65 | 9.09 | 71.64 |
> | **LFQA** | 20.89 | 32.25 | 21.34 | 10.77 | 35.85 | 11.45 | 56.44 | 48.39 | 44.06 | 22.73 | 31.34 |
> | **REFER** | 51.20 | 61.75 | 64.03 | 26.15 | 90.57 | 75.16 | 76.07 | 74.19 | 50.34 | 9.09 | 74.63 |
>
> **Mistral-7b-v0.2 Multi-IF result**
> | Models | Base Model | Turn-1 | Turn-2 | Turn-3 |
> | :--- | :--- | :---: | :---: | :---: |
> | **Vanilla** | Mistral-7B-Instruct-v0.2 | 46.06 | 33.05 | 26.03 |
> | **LFQA** | Mistral-7B-Instruct-v0.2 | 21.30 | 13.37 | 12.73 |
> | **REFER** | Mistral-7B-Instruct-v0.2 | **47.36** | **33.79** | **26.23** |
>
> **Mistral-7b-v0.2 Livebench result**
> | Models | Base Model | Paraphrase | Simplify | StoryGeneration | Summarize |
> | :--- | :--- | :---: | :---: | :---: | :---: |
> | Vanilla | Mistral-7B-Instruct-v0.2 | 37.85 | **49.97** | 42.47 | **37.23** |
> | LFQA | Mistral-7B-Instruct-v0.2 | 24.23 | 20.62 | 23.15 | 23.52 |
> | REFER | Mistral-7B-Instruct-v0.2 | **38.95** | 48.77 | **44.13** | 35.62 |
>
> **Narrow domain coverage:** We selected the Finance and Lifestyle subsets as they feature the longest average answer lengths and focused on complex, process-oriented explanations over simple factoids. This makes them a more suitable benchmark for measuring out-of-domain performance. Moreover, the core mechanism of REFER is structural, not semantic. Since the injected constraints target formatting and organization rather than domain-specific facts, the instruction-following gains are domain-agnostic and expected to transfer effectively to other specialized fields.
>
> **Question**
> **Why we need Reverse Constraint Generation:** We employ reverse generation to guarantee that the constraints are compatible with the ground truth. If we were to randomly assign a constraints, we would be forced to rewrite the answer, risking semantic drift or hallucination. By deriving constraints from the answer's existing structure (Reverse Generation), we ensure that the model learns strict format adherence while keeping the original textual content and semantics completely intact.
>
> Furthermore, our Structural Rewrite module operates on the same principle of semantic preservation. We leverage LLMs to diversify the answer structure (e.g., splitting text into paragraphs) only when it naturally improves clarity and flow, rather than forcing unnecessary restructuring that might compromise the original meaning.

---

### Official Review · Reviewer_WMcq · 2025-11-02

**Soundness:** 2
**Presentation:** 3
**Contribution:** 2
**Rating:** 4
**Confidence:** 4

**Summary:**

The paper proposes REFER, a data-centric pipeline that converts human-annotated long-form QA (LFQA) into multi-turn instruction-tuning data with verifiable format constraints.

The pipeline consists of 4 steps:
1) question rewrites to diversify prompts,
2) structural rewrites of answers,
3) feature extraction from answers using Qwen-32B + SpaCy + rules, and
4) reverse constraint generation that injects format constraints without altering semantics.

The method aims to improve instruction-following while retaining domain knowledge, addressing the finding that naive LFQA fine-tuning harms instruction following. Experiments on IFEval, Multi-IF, and LiveBench, plus a domain-adaptation study on LFRQA (Finance, Lifestyle), show Llama-2-7B fine-tuned with REFER outperforms LFQA fine-tuning and several data-augmentating baselines on constraint following and maintains competitive performance elsewhere. The paper provides prompt templates, constraint examples, training settings, and authors promised to open-source the framework and constraint pool.

**Strengths:**

- The method is clearly defined, well motivated and consist of concrete prompt templates and examples.
- The authors demonstrate the relevance of the problem, showing that naive LFQA SFT hurts instruction following, while their method REFER mitigates it. They present significant gains against compared baselines.
- Reproducibility details are provided: dataset composition, LoRA settings, training schedule, and explicit benchmark protocols are listed; prompts for augmentation are provided in the appendix. The authors promised to open-source the framework.
- Method is practical and shows effectiveness using open-source models/tools (Qwen-32B, SpaCy), training with LoRA, not needing external APIs / proprietary models for augmentation.

**Weaknesses:**

- **Limited ablations on REFER design:** It is hard to attribute where gains primarily come from. We don’t see isolation of:
  - impact of question rewrite vs structural rewrite vs reverse constraints;
  - impact of multi-turn formatting;
  - a broad discussion on sensitivity to constraint categories (length/case/punctuation/etc.,).
- **Novelty:** I am not fully convinced about the novelty of the proposed solution. There are already several lines of work that (i) turn long or structured text into instructions [1], (ii) build constraint-rich instruction-following datasets from existing content [2,3,4], (iii) fine-tune models on domain data while preserving knowledge [5], and (iv) decompose and verify multi-constraint instructions in an agent-like way at inference time [6,7]. The paper should explain more clearly what is actually new beyond combining these ideas, and whether the contribution is more than smart engineering or pipeline design.

References:

[1] Köksal, Abdullatif, et al. "LongForm: Effective Instruction Tuning with Reverse Instructions." EMNLP 2024

[2] Sun, Haoran, et al. "Conifer: Improving complex constrained instruction-following ability of large language models." arXiv preprint arXiv:2404.02823 (2024).

[3] An, Kaikai, et al. "UltraIF: Advancing Instruction Following from the Wild." arXiv preprint arXiv:2502.04153 (2025)

[4] He, Qianyu, et al. "From Complex to Simple: Enhancing Multi-Constraint Complex Instruction Following Ability of Large Language Models." EMNLP 2024.

[5] Li, Haiyun, et al. "KEFT: Knowledge-Enhanced Fine-Tuning for Large Language Models in Domain-Specific Question Answering." Transactions of the Association for Computational Linguistics (2025).

[6] Ferraz, Thomas, et al. "LLM Self-Correction with DeCRIM: Decompose, Critique, and Refine for Enhanced Following of Instructions with Multiple Constraints." EMNLP 2024.

[7] Zhang, Xianren, et al. "Divide-Verify-Refine: Aligning LLM Responses with Complex Instructions." arXiv:2410.12207 (2024)

**Questions:**

1) Why Table misses 2 “Llama-2-7B (Base)” line?

2) The authors presented the code in the supplemental material. Do the authors plan to release code? In the conclusion you promised to release the framework, but what exactly will be included in this release?

3) Can you discuss more the degradation that happens on LiveBench? How can we avoid it? Which elements of the method make it present a smaller degradation?

4) Typos:
- On line 249 “he” instead of “The”.
- Duplicate “answer answer” (line 123)

---

> ### Author Response · Authors · 2025-11-27
> **Detailed explanation and new experiments to address reviewer concerns and questions.**
>
> We sincerely thank for the insightful feedback. We value the suggestion to deepen the ablation study and have conducted additional experiments during the rebuttal phase to address these points.
>
> **Weakness: Limited ablations on REFER design**
>
> **Impact of question rewrite vs structural rewrite vs reverse constraints:** The components of REFER are designed to be interdependent. The Structural Rewrite is a prerequisite; without first restructuring the raw LFQA answer into a format that supports constraints, valid Reverse Constraints cannot be generated. We will provide a more detailed result from IFEval in our final version which shows that structural rewriting significantly improves the model's control over output structure compared to the naive LFQA baseline. Furthermore, the high semantic preservation scores in our paper (ROUGE/BERTScore) demonstrate that the Reverse Constraint generation successfully injects verifiable constraints without altering the ground-truth factual content.
>
> **Impact of Multi-turn Formatting (New Experiment):** To isolate the impact of our multi-turn design, we transformed the final round of our REFER dataset into a standard single-turn Alpaca-style instruction tuning format and fine-tuned the same model using the same hyper-parameters. Result : The REFER model outperformed Alpaca model on instruction-following benchmarks. This experiment shows that the model benefits from multi-turn QA fine-tuning.
>
> **Table 1: IFEval (Part A)**
> | Models | ChangeCase | Combination | Content | Format | Keywords | Language |
> | :--- | :---: | :---: | :---: | :---: | :---: | :---: |
> | Llama-2-7B (Alpaca) | 31.50 | 15.40 | 60.40 | 59.90 | 55.80 | 61.30 |
> | Llama-2-7B (REFER) | **62.92** | **27.68** | 60.38 | **67.52** | 47.85 | **70.97** |
>
> **Table 2: IFEval (Part B)**
> | Models | Length | Punctuation | Startend | I-Level | C-Level |
> | :--- | :---: | :---: | :---: | :---: | :---: |
> | Llama-2-7B (Alpaca) | 41.30 | 15.20 | 34.30 | 32.5 | 43.90 |
> | Llama-2-7B (REFER) | **42.66** | **54.55** | **73.13** | **43.25** | **54.92** |
>
> **Table 3: Multi-IF**
> | Models | Base Model | Turn-1 | Turn-2 | Turn-3 |
> | :--- | :--- | :---: | :---: | :---: |
> | Llama-2-7B (Alpaca) | Llama-2-7B-chat | 32.35 | 23.90 | 19.44 |
> | Llama-2-7B (REFER) | Llama-2-7B-chat | **42.09** | **30.04** | **24.47** |
>
> **Novelty and Comparison to Prior Work**
>
> We clarify how REFER differs fundamentally from the cited works ([1]-[7]):
>
> **Privacy and Open Source ([1][2][4][5][6][7]):** Our framework demonstrates that by combining NLP tools with open-source teacher models, we can eliminate reliance on external APIs. This approach ensures data security and reproducibility, making it highly viable for environments with strict enterprise policies.
>
> **Real-world dataset refinement vs. Synthetic Dataset generation ([3][6][7]):** The goal of our method is to inject new knowledge into the model while maintaining or even improving its inherent instruction-following performance. Fundamentally, our pipeline is designed to transform existing LFQA datasets rather than generating new ones from scratch. In contrast, methods like [3], [6], and [7] rely on the reasoning capabilities of LLMs to generate entirely new synthetic datasets to align the model’s instruction capabilities.
>
> **Comparison to KEFT [5]:** We note that [5] is concurrent work. While we share the goal of knowledge injection, [5] utilizes a specific loss function for NLG. In contrast, REFER utilizes a data-centric multi-turn instruction tuning approach focused specifically on verifiable format constraints, not just general reasoning ability.
>
> **Question**
>
> **Reproducibility and Contributions:** We are committed to open-sourcing the full framework. The release will include the end-to-end pipeline allowing the user to input raw LFQA data and output multi-turn instruction datasets. We will also release the detailed code to fine-tune the model with deepspeed+LoRA.
>
> **LiveBench Degradation (new experiment):** We analyzed the LiveBench degradation and identified two factors:
> * **Context Window:** Llama-2 context window is often exceeded by LiveBench prompts, leading to truncation and inconsistent evaluation.
> * **Model Scale:** To address this, we fine-tune Mistral-7B-v0.2 (which has a larger context window) using the same dataset generated by our REFER framework. The Mistral-REFER model showed performance gains on LiveBench compared to the base model.
>
> | Models | Base Model | Paraphrase | Simplify | StoryGeneration | Summarize |
> | :--- | :--- | :---: | :---: | :---: | :---: |
> | Vanilla | Mistral-7B-Instruct-v0.2 | 37.85 | **49.97** | 42.47 | **37.23** |
> | Mistral-7B (LFQA) | Mistral-7B-Instruct-v0.2 | 24.23 | 20.62 | 23.15 | 23.52 |
> | Mistral (REFER) | Mistral-7B-Instruct-v0.2 | **38.95** | 48.77 | **44.13** | 35.62 |

---

### Meta-Review · Area_Chair_zwY9 · 2026-01-06

**Summary:**

Reviewers raised interconnected concerns about experimental validity, novelty, and methodological clarity. Reviewer HDNy flagged serious discrepancies between reported LiveBench scores and the original benchmark paper’s figures, undermining confidence in empirical claims. Multiple reviewers noted insufficient ablations to isolate contributions from question rewriting, structural rewriting, and reverse constraint generation. The novelty was questioned as combining existing techniques (reverse instruction generation, constraint-based tuning, domain adaptation) without clearly articulating what’s fundamentally new beyond pipeline engineering.

**Reviewer Concerns:**

The authors conducted new Mistral-7B-v0.2 experiments demonstrating REFER’s benefits transfer to newer architectures, with LFQA fine-tuning degrading performance while REFER maintains instruction-following metrics. They clarified that LiveBench results correspond to the Instruction Following subset specifically, not the global average. A new ablation comparing multi-turn REFER against single-turn Alpaca-style formats showed meaningful gains from the multi-turn approach. Technical clarifications about constraint generation (hard vs. soft constraints, the hybrid SpaCy/LLM/rule-based pipeline) addressed some methodological opacity.

Several issues remain unresolved. Component-level ablations isolating each step’s contribution are still missing. Reviewer 8P91’s question about whether mixed training (LFQA + instruction data) achieves similar results with less complexity wasn’t addressed experimentally. Concerns about test data leakage in domain evaluation weren’t fully clarified despite mentions of GPT-4 rewrites. Narrow domain coverage limits generalizability claims.

**Reviewer Scores:**

Reviewers are likely maintain the scores

---

### Decision · Program_Chairs · 2026-01-26

Reject